# Abdomen Multi-Organ Segmentation Using Pseudo Labels and Two-Stage

Yang Xinye[0000−0002−9525−1405], Xuru Zhang[0000−0002−3916−8105], Yan Xiaochao[0000−0002−4358−6588], Ding Wangbin[0000−0001−6494−5554], Chen Hao[0000−0002−6849−9413], and Huang Liqin✉[0000−0001−8602−6380]

College of Physics and Information Engineering, Fuzhou University, Fuzhou, China
`hlq@fzu.edu.cn`

**Abstract.** Recently, the nnU-Net network had achieved excellent performance in many medical image segmentation tasks. However, it also had some obvious problems, such as being able to only perform fully supervised tasks, excessive resource consumption in the predict. Therefore, in the abdominal multi-organ challenge of FLARE23, only incomplete labeled data was provided, and the size of them was too large, which made the original nnU-Net difficult to run. Based on this, we had designed a framework that utilized generated pseudo labels and two-stage segmentation for fast and effective prediction. Specifically, we designed three nnU-Net, one for generating high-quality pseudo labels for unlabeled data, the other for generating coarse segmentation to guide cropping, and the third for achieving effective segmentation. Our method achieved an average DSC score of 88.87% and 38.00% for the organs and lesions on the validation set and the average running time and area under GPU memory-time cure are 45s and 3000MB, respectively.

**Keywords:** Pseudo Label · Two-stage · Low Consumption.

## 1 Introduction

The segmentation of abdomen organs plays an important role in the field of medical imaging. Abdomen organs are common cancer sites, including colorectal and pancreatic cancer, which are the second and third major causes of cancer deaths worldwide. Therefore, accurate segmentation of abdominal CT images is crucial for early diagnosis, treatment planning, and efficacy evaluation of cancer. Since the advent of CT(computer tomography ), it has been frequently used for the treatment and monitoring of cancer. Through CT scanning, doctors can obtain detailed three-dimensional images of the patient's internal organs in a non-invasive manner, thereby helping doctors locate and develop treatment plans. These tasks all rely on the accurate segmentation of abdominal organs. For example, in the process of cancer treatment, doctors need to quantitatively evaluate the volume changes of lesions to monitor efficacy, which requires precise lesions segmentation. In addition, organ segmentation before surgery can help doctors plan surgical paths and predict surgical risks, thereby improving the success rate of surgery.

Traditional manual segmentation methods have many limitations. This manual evaluation is subjective, resulting in significant differences between experts, and low consistency of results. Secondly, the manual segmentation process consumes a lot of time and labor, especially for large-scale datasets, which reduces work efficiency. In addition, manual segmentation may face many difficulties for complex anatomical structures and lesions, resulting in unsatisfactory segmentation results. Therefore, researchers have proposed some traditional abdominal segmentation methods, such as region growth algorithm, graph cut, morphological, and level set method, which have achieved good results. However, due to its reliance on manually designed features and rules, it has poor segmentation performance for complex organ structures and lesions and slow speed for processing large-scale data.

The introduction of deep learning technology has brought new hope for abdominal organ segmentation. Deep learning models can automatically learn the features in images, and accurately segment abdominal organs and lesions at the pixel level, greatly improving the efficiency and accuracy of segmentation. Therefore, deep learning technology is widely used in the segmentation of abdominal CT images and has achieved remarkable results in many studies. For example, the U-Net model adopts an encoding decoding structure, which effectively captures features at different scales, significantly improving the segmentation effect of organs and lesions [17]. The Seg-Net model adopts a lightweight encoding decoding structure, which is suitable for low computational resource scenarios while maintaining good segmentation accuracy [1]. Cao et al. proposed a network based on self-attention mechanism for abdominal organ segmentation [3], which can capture long-distance dependencies of images and improve generalization ability through self-supervised pre-training. This method has also achieved good results in multi-organ and tumor segmentation tasks. However, it requires a large amount of computing resources and time to train the model, which may not be feasible in practical applications. Chen et al. proposed a deep network based on incremental learning [22], which can recall old knowledge without saving old data and dynamically extends to new categories. It also utilizes visual semantic information embedded in text to enhance training effectiveness. It demonstrates superior performance in multi-organ and tumor segmentation tasks. However, as to this method, it is necessary to design a reasonable pseudo label generation strategy and parameter-sharing mechanism to alleviate catastrophic forgetting problems, select appropriate text descriptions and embedding methods to extract effective visual semantic information, balance the learning rate and weight between new and old categories to avoid overfitting or underfitting problems.

One of the most important and well-performing baselines among these methods is nnU-Net [10], namely no-new-Net. It can automatically configure parameters and conduct network training based on data. The nnU-Net places more emphasis on image preprocessing, automatically determining image modality and performing corresponding normalization operations, and resampling different voxel intervals based on cubic spline interpolation. and it can automatically set hyperparameters, such as training batch size, image block size, downsampling

frequency, etc. In recent years, many top-level solutions have been established based on it to address the challenges of medical image segmentation. Although nnU-Net can achieve state-of-the-art performance in a fully supervised manner, it is also limited to fully supervised training. Faced with complex datasets, it is difficult for nnU-Net to achieve the expected results. Generally speaking, it can be observed that nnU-Net has the following problems:

– The default nnU-Net has a high computational complexity, which takes a long time to perform preprocessing or prediction. At the same time, the device is prone to memory overflow;
– The default nnU-Net does not support training other than fully supervised.

However, in real clinical scenarios, the time budget inferred by the model and the amount of labeled data is limited. Therefore, we urgently need a framework that can utilize all types of data and perform effective inference simultaneously.

Inspired by [24] [21], in this article, we designed a two-stage framework consisting of three 3D nnU-Net. Firstly, by using incomplete labeled data to generate pseudo labels and overlaying existing labels to generate more reliable pseudo labels which serve as the basis for fully supervised data in subsequent stages; Secondly, in the first stage, coarse segmentation is performed to generate the abdominal ROI region, guiding the data crop in the second stage, thereby reducing the initial size of the data. Thirdly, perform final fully supervised fine segmentation.

Our main contributions are summarized as follows:

– We have designed a simple pseudo-labels generation framework based on nnU-Net;
– We propose an effective cropping strategy that utilizes coarse segmentation results to locate and crop ROI regions. This strategy can greatly reduce the initial data size and is beneficial for improving inference speed and reducing resource consumption.

## 2 Method

### 2.1 Preprocessing

We analyzed the original labeled data and found that some data contained partial organ labels, some data only contained tumor labeling, and some data contained both organ and tumor labeling. Due to the fact that no label in the original data contained both the tumor and all abdominal organs, we extracted the data of labels containing all organs (a total of 13 categories) as the first category data. As to the remaining data which had incomplete organ labels, we continued to divide it into two parts, one of which contained both organs and tumors as the second category. In subsequent strategies, we would use these two types of data in sequence.

We didn't use unlabeled images and the pseudo labels generated by the FLARE22 winning algorithm [9] and the best-accuracy-algorithm [19].

## 2.2   Proposed Method

As shown in the Figure 1, this is the overall framework of our method. First, use the first category data which has 13 organ-labels for training nnU-Net-14 to generate pseudo labels for the organs(According to the number of categories required in the corresponding segmentation task, we add number-suffixes to the "nnU-Net" used to distinguish different nnU-Net we used. For example, if the task has 13 labels, the number of categories is 14 and it is named nnU-Net-14. The latter two nnU-Net are also the same). Second, the pseudo labels are added to the second category data to obtain a more reliable pseudo labels. Third, unify all the pseudo labels obtained in the second step into similar labels, which used for nnU-Net-2 training for coarse segmentation of background and abdominal regions. Finally, under the guidance of nnU-Net-2 coarse segmentation, the pseudo labels got in second step with its' corresponding data are cropped and used for nnU-Net-15 training to obtain a network that can segment tumors and all organs.

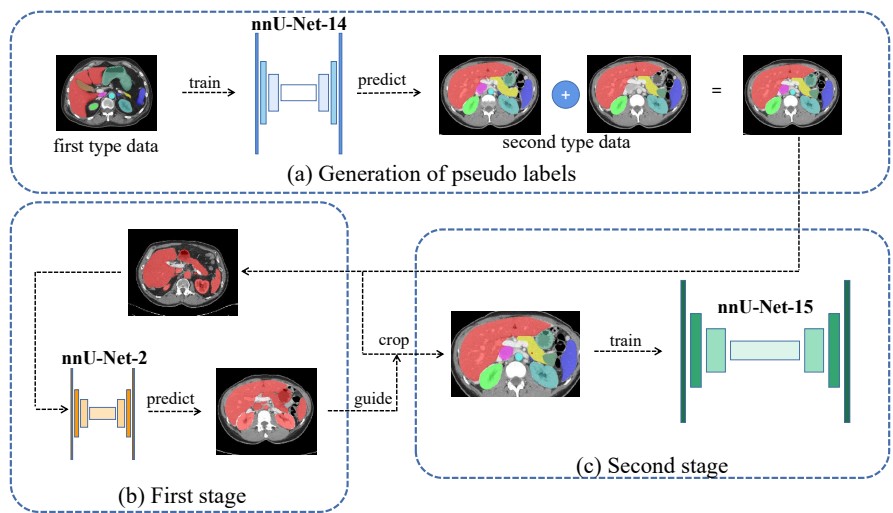

**Fig. 1.** This is the overall framework of our method. Part (a) is pseudo label generation, (b) is the first stage coarse segmentation, and (c) is the second stage fine segmentation.

**Generation of Pseudo Labels** Previously, we had already divided the dataset into several parts. For the first type of data, we used threshold algorithms to roughly remove the areas outside the torso to reduce excess areas and training costs. Firstly, the first type of data containing 13 organ labels after processing was used for fully supervised training of the first nnU-Net-14 network. Secondly, the network was used to predict the second type of data and generate

corresponding organ pseudo labels. Thirdly, used the generated pseudo labels to guide the filling of real labels, forming 14-type labels including all organs and tumors, like Figure 2. This process could also use data that only contain tumor label to be filled by predicted pseudo organ labels, getting 14-type labels. But we didn't choose it. This was because the quality of real labels was better than that of pseudo labels, the more real organ labels there were during the process, the better the quality of the final pseudo labels.

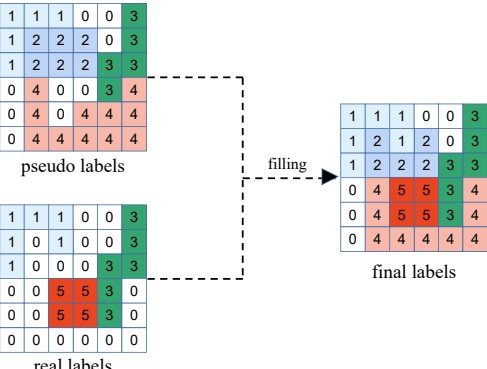

**Fig. 2.** We introduce our label-filling strategy. As shown in the figure, first identify which types of real labels are missing compared to the generated pseudo labels. In our schematic, it is obvious that the real labels do not contain two types of labels: 2 and 4. Then, we will only fill in these types of labels based on the real labels. When filling in, first determine whether the locaiton already has a label. If not, fill it in. Otherwise not to avoid overwriting the original real labels.

**First Stage Segmentation** We converted the final labels of these 14 categories into labels with only one category, and reduced them and the corresponding data's size to 1/4 of the original size to lower the consumption of training and predict processes. We used the reduced data and labels to train the nnU-Net-2 network and used it to predict any reduced data to obtain simple abdominal region segmentation results. Finally, by expanding the segmentation result to the initial size, we obtained the final coarse segmentation result.

**Second Stage Segmentation** We could use the first stage coarse segmentation network to predict the second type of data, and guided cropping for the second type of data based on the abdominal region labels in the coarse segmentation results. Due to the high amount of impurities in the coarse segmentation results, there were many misclassified areas, and the abdominal organs'label did not

stick together, these factors would affect our strategy of finding the abdominal range. Therefore, we first dilated the labels of the coarse segmentation results to make the abdominal organs as connected as possible. Then counted all connected domains, among which the largest connected domain was basically the abdominal region. We took its range to guide subsequent data cropping and used these data for the training of nnU-Net-15.

### 2.3   Our Method Used in Inference

In inference, we first reduced the input data size to 1/4. Then used nnU-Net-2 for the first prediction on the reduced data to obtain the coarse segmentation result, then restored its size. Secondly, we got the spatial range of the largest connected domain in the coarse segmentation results, which was very likely abdomen, to guide the cropping of the original validation data. Finally, nnU-Net-15 was used to predict the validation data after the cropping, obtaining a fine segmentation. The size of the fine segmentation result was filled to the same size as the original data to obtain the final segmentation result. Although our method used two predictions, it did not take more time than end-to-end nnU-Net. Moreover, we used different methods before each stage to greatly reduce the size of input data, resulting in improved inference speed and lower resource consumption in each stage.

## 3   Experiments

### 3.1   Dataset and evaluation measures

The FLARE 2023 challenge is an extension of the FLARE 2021-2022 [13][14], aiming to aim to promote the development of foundation models in abdominal disease analysis. The segmentation targets cover 13 organs and various abdominal lesions. The training dataset is curated from more than 30 medical centers under the license permission, including TCIA [4], LiTS [2], MSD [18], KiTS [7,8], autoPET [6,5], TotalSegmentator [20], and AbdomenCT-1K [15]. The training set includes 4000 abdomen CT scans where 2200 CT scans with partial labels and 1800 CT scans without labels. The validation and testing sets include 100 and 400 CT scans, respectively, which cover various abdominal cancer types, such as liver cancer, kidney cancer, pancreas cancer, colon cancer, gastric cancer, and so on. The organ annotation process used ITK-SNAP [23], nnU-Net [11], and MedSAM [12].

The evaluation metrics encompass two accuracy measures—Dice Similarity Coefficient (DSC) and Normalized Surface Dice (NSD)—alongside two efficiency measures—running time and area under the GPU memory-time curve. These metrics collectively contribute to the ranking computation. Furthermore, the running time and GPU memory consumption are considered within tolerances of 15 seconds and 4 GB, respectively.

### 3.2  Implementation details

**Environment settings**  The development environments and requirements are presented in Table 1.

**Table 1.** Development environments and requirements.

| | |
|---|---|
| System | Ubuntu 20.04.2 |
| CPU | Intel(R) Core(TM) i9-12900X CPU@3.13GHz |
| RAM | 16×4GB/s |
| GPU (number and type) | NVIDIA GeForce RTX 4090 24G |
| CUDA version | 11.7 |
| Programming language | Python 3.9 |
| Deep learning framework | Pytorch (Torch 2.0.1) |

**Training Protocols**  The training protocols for three nnU-Net are shown in Table 2.

**Table 2.** Training protocols.

| | Labels generation | Stage 1 | Stage 2 |
|---|---|---|---|
| Batch size | | 2 | |
| Initial learning rate (lr) | | 0.01 | |
| Lr decay schedule | | polylrscheduler for nnU-Net | |
| Patch size | 64×128×224 | 96×160×160 | 112×128×160 |
| Total epochs | 1000 | 50 | 500 |
| Training time | 36 hours | 2hours | 14hours |
| Optimizer | | SGD with nesterov momentum ($\mu = 0.99$) | |
| Loss | | RobustCrossEntroyLoss + MeomoryEfficientSoftDiceLoss | |

## 4  Results and discussion

### 4.1  Quantitative results on validation set

The performance of trained nnU-Net-14 and nnU-Net-15 on the validation is shown in the table 3. It can be seen that nnU-Net-15, trained after label filling, performs better on most organ segmentation in the validation than nnU-Net-14 which only segments organs.

**Table 3.** Quantitative evaluation results.

| Target | nnU-Net-14 | | nnU-Net-15 | |
|---|---|---|---|---|
| | DSC(%) | NSD(%) | DSC(%) | NSD(%) |
| Liver | 0.9363 | 0.9539 | 0.9663 | 0.9599 |
| Right Kidney | 0.9032 | 0.9168 | 0.9229 | 0.9204 |
| Spleen | 0.9121 | 0.9169 | 0.9673 | 0.9658 |
| Pancreas | 0.8062 | 0.9365 | 0.8445 | 0.9462 |
| Aorta | 0.9439 | 0.9676 | 0.9558 | 0.9744 |
| Inferior vena cava | 0.9171 | 0.9303 | 0.9323 | 0.9435 |
| Right adrenal gland | 0.7938 | 0.9275 | 0.8284 | 0.9457 |
| Left adrenal gland | 0.7676 | 0.9008 | 0.7853 | 0.9117 |
| Gallbladder | 0.7471 | 0.7159 | 0.7849 | 0.7721 |
| Esophagus | 0.8063 | 0.9175 | 0.8146 | 0.9218 |
| Stomach | 0.8895 | 0.9305 | 0.9003 | 0.9346 |
| Duodenum | 0.8895 | 0.9199 | 0.8269 | 0.9411 |
| Left kidney | 0.8956 | 0.8943 | 0.9240 | 0.9196 |
| Tumor | / | / | 0.3914 | 0.3286 |
| Average | 0.8547 | 0.9100 | 0.8810 | 0.9274 |

### 4.2   Qualitative results on validation set

In the two cases with the best segmentation performance, like Figure 3, it can be seen that the organ and tumor are well segmented. In the two cases with the worst segmentation performance, we found that our model's prediction results clearly had regular boundaries, causing the segmentation results to appear as if some parts have been removed, as shown in the red box in the Figure 4.

### 4.3   Segmentation efficiency results on validation set

After observation, most of the data show similar utilization rates in CPU, GPU, and RAM, so we randomly select two of them for analysis, like Figure 5. Because this is a two-stage segmentation network, there are two prediction stages, which will occupy a lot of RAM and GPU. Therefore, there will be two peaks in the variation of GPU and RAM occupancy over time. The first smaller peak is due to the coarse segmentation, while the second larger peak is due to the fine segmentation in the second stage.

### 4.4   Results on final testing set

The table 5 is our testing results during MICCAI (2023.10.8).

### 4.5   Limitation and future work

For limitation, from examples of poor segmentation, we can easily conclude that the segmentation performance of our method not only depends on the quality

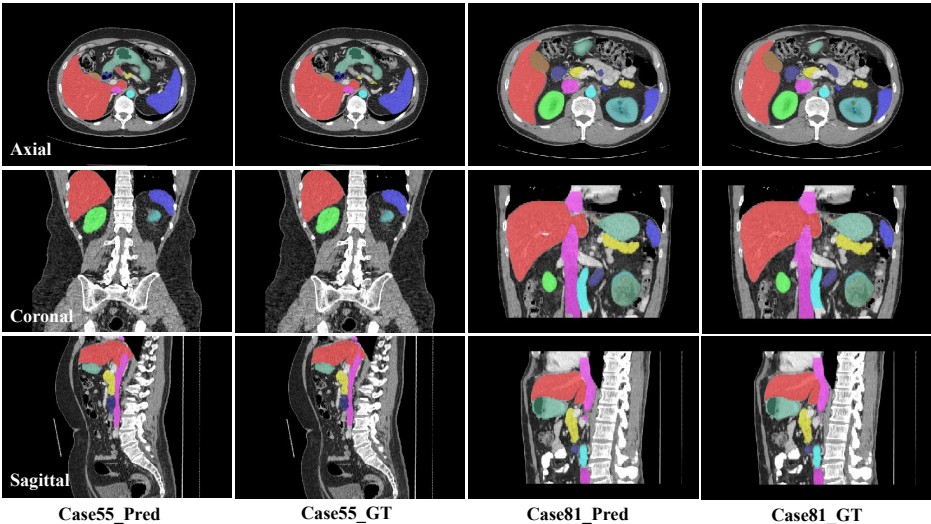

**Fig. 3.** Two best examples from our segmentation results.

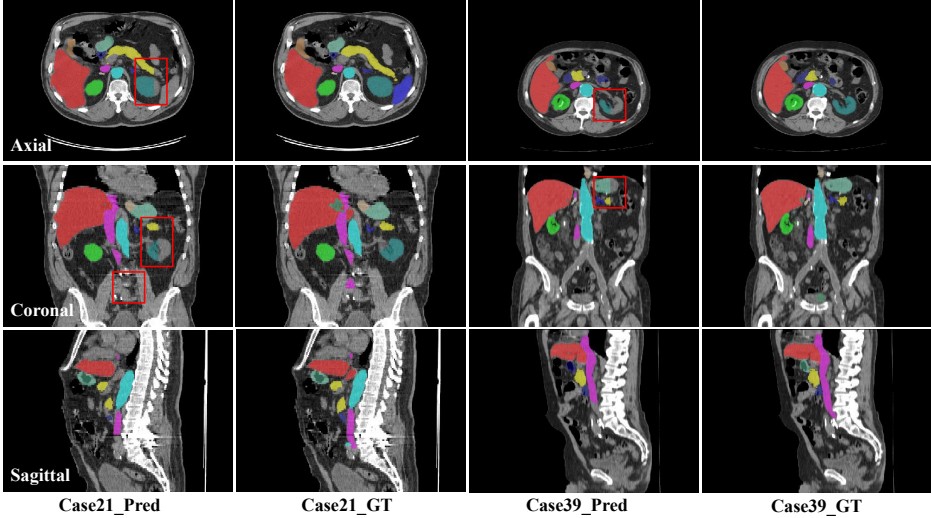

**Fig. 4.** Two worst examples from our segmentation results.

**Table 4.** Quantitative evaluation of segmentation efficiency in terms of the running them and GPU memory consumption.

| Case ID | Image Size | Running Time (s) | Max GPU (MB) | Total GPU (MB) |
|---------|------------|------------------|--------------|----------------|
| 0001 | (512, 512, 55) | 93.76 | 3882 | 16384 |
| 0051 | (512, 512, 100) | 107.06 | 4174 | 16384 |
| 0017 | (512, 512, 150) | 127.36 | 4394 | 16384 |
| 0019 | (512, 512, 215) | 111.72 | 3946 | 16384 |
| 0099 | (512, 512, 334) | 129.2 | 3848 | 16384 |
| 0063 | (512, 512, 448) | 162.63 | 3922 | 16384 |
| 0048 | (512, 512, 499) | 173.74 | 3794 | 16384 |
| 0029 | (512, 512, 554) | 225.18 | 4554 | 16384 |

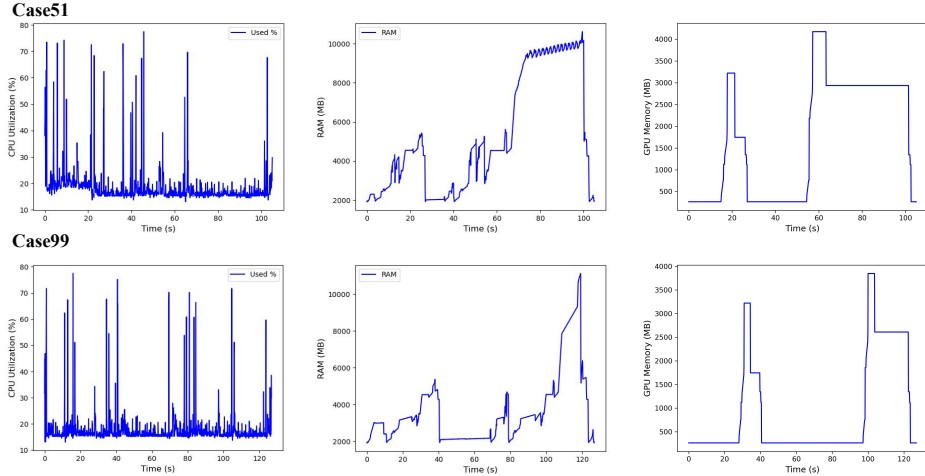

**Fig. 5.** Qualitative evaluation of segmentation efficiency.

**Table 5.** Results on final testing set

| Accuracy measures | DSC | NSD |
|---|---|---|
| Liver | 0.9646 | 0.9656 |
| RK | 0.9447 | 0.9423 |
| Spleen | 0.9252 | 0.9311 |
| Pancreas | 0.8925 | 0.9659 |
| Aorta | 0.9672 | 0.9842 |
| IVC | 0.9540 | 0.9704 |
| RAG | 0.8305 | 0.9445 |
| LAG | 0.8014 | 0.9270 |
| Gallbladder | 0.7894 | 0.8033 |
| Esophagus | 0.8743 | 0.9690 |
| Stomach | 0.9074 | 0.9505 |
| Duodenum | 0.8610 | 0.9561 |
| LK | 0.9127 | 0.9168 |
| Organ | 0.8940 | 0.9407 |
| Lesion | 0.3379 | 0.2474 |

of the pseudo labels, but also on the accuracy of the first stage coarse segmentation. Poor quality of coarse segmentation can lead to incomplete coverage to the abdominal area after cropping, resulting in segmentation fragmentary. For future work, on the one hand, we plan to adopt the idea of iterative learning and continuously utilize better models to generate more accurate pseudo labels; On the other hand, we plan to improve the performance of the coarse segmentation network and improve the post-processing strategy of it.

## 5    Conclusion

This article designs a two-stage segmentation framework based on nnU-Net, which utilizes partially labeled data for fully supervised data construction, training, and effective predict. This method can solve the problem of excessive consumption of nnU-Net.

**Acknowledgements** The authors of this paper declare that the segmentation method they implemented for participation in the FLARE 2023 challenge has not used any pre-trained models nor additional datasets other than those provided by the organizers. The proposed solution is fully automatic without any manual intervention. We thank all the data owners for making the CT scans publicly available and CodaLab [16] for hosting the challenge platform.

This work was supported by National Natural Science Foundation of China (62271149), Fujian Provincial Natural Science Foundation project(2021J02019).

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

**Table 6.** Checklist Table. Please fill out this checklist table in the answer column.

| Requirements | Answer |
| --- | --- |
| A meaningful title | Yes |
| The number of authors ($\leq$6) | 6 |
| Author affiliations and ORCID | Yes |
| Corresponding author email is presented | Yes |
| Validation scores are presented in the abstract | Yes |
| Introduction includes at least three parts: background, related work, and motivation | Yes |
| A pipeline/network figure is provided | Figure 1 |
| Pre-processing | Page 3 |
| Strategies to use the partial label | Page 3,4 |
| Strategies to use the unlabeled images. | No |
| Strategies to improve model inference | 6 |
| Post-processing | No |
| Dataset and evaluation metric section is presented | Page 6 |
| Environment setting table is provided | Table 1 |
| Training protocol table is provided | Table 2 |
| Ablation study | Page 7 |
| Efficiency evaluation results are provided | Table 4 |
| Visualized segmentation example is provided | Figure 3,4 |
| Limitation and future work are presented | Yes |
| Reference format is consistent. | Yes |