# OpenReview forum: "Abdomen Multi-Organ Segmentation Using Pseudo Labels and Two-Stage"
_MICCAI.org/2023/FLARE — Submitted to FLARE 2023_

### Official Review · Reviewer_uBXC · 2023-09-19
**good job**

**Rating:** 6
**Confidence:** 4

**Review:**

Pros:
1. This paper contains necessary parts.
2.  The solution is clear

Cons:
1. Missing the post-processing and strategies for accelaration
2. Fig 5 has low resolution
3. More details of experiment should be presented, eg. ablation study.

---

> ### Comment · Reviewer_uBXC · 2023-11-20
>
> Cons:
> 1. The Fig 5 is still unclear
> 2. The section 4.4 should be modified.

---

> > ### Author Response · Authors · 2023-11-22
> > **Adout4.4**
> >
> > Excuse me, what should I modify in section 4.4?

---

### Official Review · Reviewer_7ECB · 2023-09-26

**Rating:** 8
**Confidence:** 4

**Review:**

This is a well-written paper with sufficient method demonstration and ablation experiments. This paper applies a supervised-based method nnU-Net to segment abdominal organs and tumors with weakly labeled data. The two-stage pipeline strikes a good balance between segmentation performance and computational cost.

Here are some trivial personal suggestions:

1. I'm a little confused about what the 'X' means in model 'nnU-Net-X'. My understanding is that the number denotes the number of segmentation categories in the nnU-Net. Maybe I did not see specific descriptions for it due to my omissions. If not, please add it.  Also, a similar question is about 'nnU-Net-13' in the part of 'Generation of Pseudo Labels'. Is it a typo that refers to 'nnU-Net-14 ' in Fig.1 or the fourth network in the proposed method?

2. Few typos in uppercase and lowercase or spacing in sentences I noticed, for instance:

   - "First, **Use** the first category data for nnU-Net-14 segmentation training of 13 organs to generate pseudo labels for the organs."
   - "Due to the high amount of impurities in the coarse segmentation results, there were many misclassified areas, and the abdominal **organs’label** did not stick together, ..."

3. To put legends in the top-left corner in Fig. 3~5 will keep a higher resolution so that can be seen clearer.

---

### Official Review · Reviewer_m1Q5 · 2023-10-04
**Overall, the paper presents an exciting approach to abdominal multi-organ segmentation. However, it could benefit from improved clarity.**

**Rating:** 7
**Confidence:** 4

**Review:**

Abstract:

This paper proposed a novel approach to abdominal multi-organ segmentation, addressing challenges faced by the nnU-Net network. It achieved competent and precise results by leveraging pseudo labels and a two-stage segmentation process, with an average DSC score of 88.87% for organs and 38.00% for lesions on the validation set.

Introduction:
The authors highlight the significance of accurate abdominal organ segmentation in the introduction, particularly for cancer diagnosis and treatment planning. It emphasizes the limitations of manual segmentation methods and the potential of deep learning techniques like nnU-Net for improved results.

Positives:

The authors addressed a critical problem in medical imaging with practical applications. Furthermore, using pseudo labels and a two-stage approach is innovative and could improve segmentation efficacy.

Provided quantitative and qualitative results offer a complete assessment of the proposed method's performance.
Discussion of limitations and future work validates awareness of conceivable areas for improvement.

Negatives:

The title of the paper could be improved such as "Abdomen Multi-Organ Segmentation Using Pseudo Labels and Two-Stage Approach"

This research lacks clarity and organization in some sections, making it challenging to follow the methodology and results.

The abstract could be more concise and focused on key results and contributions.

Some training protocols could be added to Table 2, such as loss function, number of model parameters, flops, etc.

Check English grammar and typos, such as "improveing" throughout your manuscript.

Overall, the paper presents an exciting approach to abdominal multi-organ segmentation. However, it could benefit from improved clarity.

---

### Official Review · Reviewer_z1Mk · 2023-10-04
**2 stage nnU-Net method**

**Rating:** 8
**Confidence:** 5

**Review:**

Pros:
Complete structure;

Cons:
1. Some important preprocessing paramters are not introduced, such as target spacing;
2. There is no ablation study for accuray or efficiency.

---

### Decision · Program_Chairs · 2023-10-24

Accept